# Potential Utility of Neurosonology in Paroxysmal Atrial Fibrillation Detection in Patients with Cryptogenic Stroke

**DOI:** 10.3390/jcm8112002

**Published:** 2019-11-16

**Authors:** Chrissoula Liantinioti, Lina Palaiodimou, Konstantinos Tympas, John Parissis, Aikaterini Theodorou, Ignatios Ikonomidis, Maria Chondrogianni, Christina Zompola, Sokratis Triantafyllou, Andromachi Roussopoulou, Odysseas Kargiotis, Aspasia Serdari, Anastasios Bonakis, Konstantinos Vadikolias, Konstantinos Voumvourakis, Leonidas Stefanis, Gerasimos Filippatos, Georgios Tsivgoulis

**Affiliations:** 1Second Department of Neurology, “Attikon” University Hospital, School of Medicine, National and Kapodistrian University of Athens, 12462 Athens, Greece; chrissa21@hotmail.com (C.L.); lina_palaiodimou@yahoo.gr (L.P.); katetheo24@gmail.com (A.T.); mariachondrogianni@hotmail.gr (M.C.); chriszompola@yahoo.gr (C.Z.); socrates_tr@hotmail.com (S.T.); an.rousso@yahoo.gr (A.R.); bonakistasos@yahoo.com (A.B.); cvoumvou@otenet.gr (K.V.); lstefanis@bioacademy.gr (L.S.); 2Second Department of Cardiology, “Attikon” University Hospital, Medical School, National and Kapodistrian University of Athens, 12462 Athens, Greece; kostas.tympas@yahoo.gr (K.T.); jparissis@yahoo.com (J.P.); ignoik@gmail.com (I.I.); geros@otenet.gr (G.F.); 3Stroke Unit, Metropolitan Hospital, 18547 Piraeus, Greece; kargiody@gmail.com; 4Department of Neurology, University Hospital of Alexandroupolis, Democritus University of Thrace, School of Medicine, 68100 Alexandroupolis, Greece; aserdari@yahoo.com (A.S.); vadikosm@yahoo.com (K.V.); 5First Department of Neurology, Eginition Hospital, National and Kapodistrian University of Athens, School of Medicine, 11528 Athens, Greece

**Keywords:** cryptogenic stroke, atrial fibrillation, neurosonology, Holter monitoring, transcranial Doppler, cervical duplex

## Abstract

Background: Occult paroxysmal atrial fibrillation (PAF) is a common and potential treatable cause of cryptogenic stroke (CS). We sought to prospectively identify independent predictors of atrial fibrillation (AF) detection in patients with CS and sinus rhythm on baseline electrocardiogram (ECG), without prior AF history. We had hypothesized that cardiac arrhythmia detection during neurosonology examinations (Carotid Duplex (CDU) and Transcranial Doppler (TCD)) may be associated with higher likelihood of AF detection. Methods: Consecutive CS patients were prospectively evaluated over a six-year period. Demographics, clinical and imaging characteristics of cerebral ischemia were documented. The presence of arrhythmia during spectral waveform analysis of CDU/TCD was recorded. Left atrial enlargement was documented during echocardiography using standard definitions. The outcome event of interest included PAF detection on outpatient 24-h Holter ECG recordings. Statistical analyses were performed using univariate and multivariate logistic regression models. Results: A total of 373 patients with CS were evaluated (mean age 60 ± 11 years, 67% men, median NIHSS-score 4 points). The rate of PAF detection of any duration on Holter ECG recordings was 11% (95% CI 8%–14%). The following three variables were independently associated with the likelihood of AF detection on 24-h Holter-ECG recordings in both multivariate analyses adjusting for potential confounders: age (OR per 10-year increase: 1.68; 95% CI: 1.19–2.37; *p* = 0.003), moderate or severe left atrial enlargement (OR: 4.81; 95% CI: 1.77–13.03; *p* = 0.002) and arrhythmia detection during neurosonology evaluations (OR: 3.09; 95% CI: 1.47–6.48; *p* = 0.003). Conclusion: Our findings underline the potential utility of neurosonology in improving the detection rate of PAF in patients with CS.

## 1. Introduction

The etiology of acute cerebral ischemia (ACI) remains undetermined in more than one-third of all ischemic stroke (IS) patients upon discharge [1,2]. According to Trial of ORG 10172 in Acute Stroke Treatment (TOAST) classification, an IS is classified as cryptogenic stroke (CS) when no cause can be identified after the baseline diagnostic workup [3]. A well-defined etiopathogenic mechanism is cardioembolism, which actually accounts for 17% to 30% of all IS, with more than half of cardioembolic strokes being attributed to atrial fibrillation (AF) [4,5,6]. However, paroxysmal AF (PAF) is frequently undetected, due to episodic and asymptomatic nature and short duration [7]. It is therefore evident that a proportion of strokes labeled as CS are cardioembolic in origin because of occult PAF [8].

The detection of PAF is of utmost importance in order to provide the most suitable treatment for stroke secondary prevention. Antiplatelet treatment, advocated by current guideline recommendations for patients with CS [9], is known to provide inadequate protection from future cardioembolic events in patients with AF [10]. On the contrary, it has been estimated that the administration of anticoagulant therapy reduces the annual IS recurrence risk by 8.4% compared with antiplatelet therapy in IS patients with AF [11]. Both ESO/AHA guidelines recommend at least 24-h Holter monitoring in patients with CS to detect PAF [9,12].

Neurovascular imaging is also essential for accurate delineation of the stroke mechanism and the development of acute stroke therapies [13]. Carotid duplex ultrasound (CDU) and transcranial doppler ultrasound (TCD) are ancillary diagnostic tests in support of the etiological workup of IS and the evaluation of neurovascular status [14,15]. Both neurosonological modalities can be performed at the bedside in the very early stages of IS and are relatively inexpensive and noninvasive. Additionally, they allow monitoring and provide actual hemodynamic information. Thus, CDU and TCD may detect heart rhythm alterations in real-time during spectral waveform analysis [16,17] and provide complementary information to 24-h Holter-ECG recordings.

In view of former considerations, we sought to identify independent predictors of AF detection in patients with CS and sinus rhythm on baseline cardiac evaluation (electrocardiogram (ECG) and 24-h Holter-ECG recordings), without prior AF history. More specifically, we had hypothesized that cardiac arrhythmia detected during neurosonology evaluation (CDU and TCD) may be associated with higher likelihood of AF detection.

## 2. Methods

Consecutive patients with CS, no prior AF history and sinus rhythm on the baseline ECG and the 24-h Holter-ECG recordings were prospectively evaluated at a tertiary care stroke center (“Attikon” University Hospital, National and Kapodistrian University of Athens, Athens, Greece) over a six-year period. CS was defined according to TOAST criteria [3], following an extensive diagnostic workup of all patients presenting with symptoms of ischemic stroke (IS) or transient ischemic attack (TIA). More specifically, all patients underwent the following laboratory and imaging examinations: brain CT-scan or MRI-scan, full blood count, biochemical blood analysis (cholesterol and glucose values included), ECG, cardiac ultrasound, 24-h Holter heart-rhythm monitoring, CDU and TCD. Additional information regarding the diagnostic workup of CS patients in our center has been previously described [18,19].

Stroke severity at hospital admission was documented using National Institute of Health Stroke Scale (NIHSS) score [20] by certified vascular neurologists [18,19]. Baseline characteristics including demographics, vascular risk factors, admission NIHSS-scores, neuroimaging and neurosonology findings, echocardiographic measurements, and number of 24-h Holter monitoring evaluations were recorded. Radiologists blinded to the patients’ clinical data analyzed neuroimaging examinations and cerebral infarctions were subsequently categorized according to their location as either cortical or non-cortical including subcortical, brainstem and cerebellar location [21].

CDU and TCD examinations were performed by a certified neurosonologist (GT) with a Refurbished Philips^®^ CX50 portable ultrasound machine, using L12-3 and S5-1 ultrasound transducer probes respectively. Neurosonology examinations were performed in each patient within 48 h from hospital admission. Irregular duration of the intervals between consecutive peak-systolic velocities during spectral waveform analysis of extra- or intra-cranial arteries in at least three complexes indicated the presence of cardiac arrhythmia and this finding was prospectively documented. (Figure 1) [16,17]. Echocardiogram was performed by certified cardiologists and left atrial (LA) diameter was measured using standardized methodology as previously described [22]. LA enlargement was classified into mild, moderate or severe, according to the guidelines of the American Society of Echocardiography (ASE) [23].

Twenty-four-hour Holter ECG was performed using a 12-channel Holter monitoring Mortara H12+™ instrument. The inpatient recordings were completed within 96 h from hospital admission. It should be emphasized that sinus rhythm on baseline 24-h Holter ECG was a prerequisite for patients’ inclusion in our study and for the diagnosis of CS [3]. During the follow-up period that varied between three to 60 months, CS patients underwent ≥1 outpatient 24-h Holter ECG recordings, based on the presence of premature atrial contractions on the baseline 24-h Holter ECG and that decision was not related to the neurosonology findings. The primary outcome event of interest included PAF detection of any duration as previously described [18]. Two blinded investigators using dedicated analysis software analyzed all ECG recordings [18]. Total time in AF was calculated as the sum of each individual AF episode for patients with multiple episodes during monitoring. The secondary outcome of interest included the current definition of PAF according to ACC/AHA/ESC guidelines, which applies to AF episodes without a reversible cause lasting >30 s [24].

The study protocol was approved by the ethics committee of our hospital and signed informed consent was obtained from the patient or legal representative before enrollment in all cases.

### Statistical Analyses

Continuous variables are presented as mean ± SD (normal distribution) and as median with interquartile range (skewed distribution). Categorical variables are presented as percentages with their corresponding 95% Confidence Intervals (95% CI). Statistical comparisons between two groups were performed using χ^2^ test, or in case of small expected frequencies, Fisher’s exact test. Continuous variables were compared by the use of the unpaired *t*-test or Mann–Whitney *U* test, as indicated. Univariable and multivariable binary logistic regression models were used to evaluate associations between baseline characteristics (demographics, vascular risk factors, stroke severity, neuroimaging and neurosonology findings, echocardiographic measurements, and number of 24-h Holter monitoring evaluations) with the likelihood of detecting AF on Holter monitoring in patients with CS before and after adjusting for potential confounders. A cut-off of *p* < 0.1 was used to select variables for inclusion in multivariable analyses that were conducted using backward stepwise selection procedure. To confirm the robustness of multivariable models, we repeated all multivariable analyses using a forward selection procedure. Associations are presented as odds ratios (OR) with corresponding 95% confidence intervals (CI). Statistical significance was achieved if the *p* value was ≤0.05 in multivariable logistic regression analyses. The Statistical Package for Social Science (SPSS Inc., Armonk, NY, USA; version 23.0 for Windows) was used for statistical analyses.

## 3. Results

A total of 373 patients with CS (mean age 60 ± 11 years, 67% men, median NIHSS score on admission: four, IQR: 3–10) underwent 24-h Holter-ECG evaluations during the six-year study period. The baseline characteristics of the study population are presented in Table 1. The mean CHA2DS2-VASC score and the mean number of outpatient 24-h Holter-ECG recordings were 3.8 ± 1.3 and 1.5 ± 1.5 respectively. Moderate or severe left atrial enlargement were present in 6% of the study population, while in 20% we detected cardiac arrhythmia during neurosonology evaluations.

AF of any duration was documented on outpatient 24-h Holter-ECG recordings in 40 patients with CS (11%, 95% CI: 8–14%). The mean duration of AF was 4940 ± 1043 s, while in 12 patients (30% of AF patients) AF duration was ≤30 s. The detection rate of AF ≥30 s was 8% (95% CI: 5–11%) in our cohort. AF detection rates differed significantly (*p* < 0.001) according to the degree of left atrial enlargement (Table 2). More specifically, the rates of AF detection were 7%, 12%, and 36% in patients with no, mild, moderate, or severe left atrial enlargement respectively (*p* for linear trend <0.001). AF detection rates also differed significantly (*p* = 0.048) according to the number of 24-h Holter-ECG recordings (Table 3). More specifically, the rates of AF detection were 8%, 14%, and 21% in patients with 1, 2, and ≥3 Holter recordings respectively (*p* for linear trend 0.014).

Further evaluation regarding the cardiac structure was also conducted in a subset of our patients. Fifty-three percent of our patients (199/373) had undergone transesophageal echocardiogram (TEE). Cardiac CT and/or cardiac MRI were performed in three cases only, since these two investigations were not readily available in our hospital. All TEE, cardiac CT, and cardiac MRI investigations did not disclose any cardiogenic source of embolization in our cohort.

The univariable and multivariable associations of baseline characteristics with the likelihood of AF detection on 24-h Holter-ECG recordings are presented in Table 4. The following variables were associated with AF detection on initial univariable analyses using a *p* value of <0.1 as threshold for inclusion in multivariable models: age (OR per 10-year increase: 1.81; 95%CI: 1.31–2.50; *p* < 0.001), heart failure (OR: 2.74; 95% CI: 0.85–8.83; *p* = 0.093), CHA2DS2-VASC score (OR per 1-point increase: 1.53; 95% CI: 1.20–1.941; *p* = 0.001), ≥3 Holter recordings (OR: 2.40; 95% CI: 0.97–5.95; *p* = 0.058), moderate or severe left atrial enlargement (OR: 5.70; 95% CI: 2.22–14.61; *p* < 0.001), and cardiac arrhythmia detection during neurosonology evaluations (OR: 3.77; 95% CI: 1.87–7.60; *p* < 0.001). The following three variables were independently (*p* < 0.05) associated with the likelihood of AF detection on 24-h Holter-ECG recordings in multivariable logistic regression analyses conducted by backward selection procedure: age (OR per 10-year increase: 1.68; 95% CI: 1.19–2.37; *p* = 0.003), moderate or severe left atrial enlargement (OR: 4.81; 95% CI: 1.77–13.03; *p* = 0.002), and cardiac arrhythmia detection during neurosonology evaluations (OR: 3.09; 95%CI: 1.47–6.48; *p* = 0.003). We repeated the multivariable analyses using the forward selection procedure and obtained identical results. The independent associations of age (OR per 10-year increase: 1.68; 95% CI: 1.19–2.37; *p* = 0.003), moderate or severe left atrial enlargement (OR: 4.81; 95%CI: 1.77–13.03; *p* = 0.002) and cardiac arrhythmia detection during neurosonology evaluations (OR: 3.09; 95% CI: 1.47–6.48; *p* = 0.003) with the likelihood of AF detection persisted also on multivariable logistic regression analyses conducted by the forward selection procedure.

## 4. Discussion

Our prospective single-center cohort study showed that detection of PAF in patients with CS is independently associated with increasing age, LA enlargement, and cardiac arrhythmia detection during neurosonology evaluations. In addition, the detection rates of AF of any duration and AF ≥ 30 s on outpatient 24-h Holter-ECG recordings were 11% and 8% respectively in our cohort.

There is mounting literature suggesting that newly diagnosed AF is identified in ≈5% of patients with stroke in the inpatient setting [25], while the rate of PAF detection in CS patients varies between 5–20%, according to different studies and prolonged Holter-ECG monitoring [26,27,28]. Repetition of 24-h Holter recording can detect AF at a higher rate, as it was also demonstrated in the present study, but it still carries lower diagnostic yield compared to continuous arrhythmia monitoring [29]. The detection of occult PAF has important therapeutic implications in CS patients, as anticoagulation is the optimal treatment for secondary stroke prevention in AF-associated stroke and can substantially reduce recurrent stroke and systemic embolism compared to antiplatelet therapy [30,31,32,33,34]. Furthermore, secondary prevention in CS includes oral anticoagulation when AF is detected, regardless of AF pattern (paroxysmal or chronic). Notably, the benefit of oral anticoagulation therapy in secondary stroke prevention in patients with AF has been established both for chronic and intermittent AF [35].

Another important finding is that AF duration was ≤30 s in 30% of the patients recognized with AF in outpatient Holter monitoring in our study, while the current American College of Cardiology/American Heart Association definition of PAF requires >30 s as a threshold for AF diagnosis [24]. However, AF of any duration should be considered clinically relevant in patients with CS, as recognized bursts of PAF may be markers of longer periods of AF that occur outside of the monitoring period. Interestingly, prior studies in CS patients have used a variety of time thresholds, ranging from 0 s to 5 minutes, reflecting the lack of consensus regarding AF duration yield [36,37,38].

Our study also disclosed an association between advancing age and detection of PAF in patients with CS, which persisted on multivariable analysis. This finding is consistent with other cohort studies, which demonstrated that older age was an independent predictor of occult PAF in CS patients [39,40].

AF detection on 24-h Holter monitoring is also associated with LA enlargement. According to our study, the rates of AF detection were 7%, 12% and 36% in patients with no, mild, moderate or severe left atrial enlargement respectively and that association was statistically significant. However, the metric used for indicating LA in our study was the LA diameter, whereas LA volume indexed to the subjects’ body surface area, which represents a three-dimensional size of the LA is thought to be a superior metric of LA dimension in terms of predicting cardiovascular outcomes [41]. A recent study outlined the association of higher LA volume index with cardioembolic stroke and the rate of AF detection in patients with embolic stroke of undetermined source (ESUS), who completed four-week outpatient cardiac event monitoring [42]. ESUS is a subtype of CS and is used to describe non-lacunar CS in which embolism is a likely underlying mechanism [43]. However, ESUS constitutes a heterogenous group of patients, in whom other embolic mechanisms (patent foramen ovale, aortic plaque, non-stenosing unstable carotid plaque, cardiac valve disorders, coagulation disorders in patients with occult cancer) might be responsible for stroke, except for occult AF. Those underlying mechanisms mandate different management than oral anticoagulation, thus clinical utility of ESUS is debatable [44]. Consequently, our findings lend support to the recent concept that LA diameter measurement may help stratify ESUS patients with the greater benefit from anticoagulation due to underlying occult AF [44].

The potential diagnostic utility of neurosonology examinations (CDU and TCD) in the early detection of PAF in patients with CS is also supported by our results. Specifically, it was shown that cardiac arrhythmia detection during spectral waveform analysis in CDU/TCD evaluations was associated with the likelihood of AF detection on outpatient 24-h Holter-ECG recordings. A plausible explanation for this association may be related to the psychological stress induced by the TCD and CDU examinations to the patients that in turn may provoke episodes of arrhythmia, thus increasing the neurosonology rates of arrhythmia detection [45,46]. Being inexpensive, readily available, performed by-the-bed in the early stages of IS, even before the first 24-h Holter recording has been completed, CDU/TCD examination can be a useful tool for delineating stroke etiology in a multifactorial approach; both evaluating extracranial/intracranial vascular stenosis or occlusion and detecting cardiac arrhythmias in real-time [15,16,17]. One limiting factor is that, although neurosonology examinations can detect arrhythmias, it is not possible to differentiate them among the many different types and provide a certain diagnosis of AF. Abnormal neurosonology examination can represent AF as simple extra-systolic beats and consequently the specificity of this examination as a predictor of AF appears low. However, AF appears to account for a substantial proportion of rhythm abnormalities [47]. Even if the cardiac arrhythmia detected by neurosonology examinations is finally diagnosed as paroxysmal supraventricular tachycardia (PSVT) in ECG studies, this is also clinically relevant information, as PSVT patients have higher prevalence rate of AF [48]. Consequently, arrhythmia detection by CDU/TCD can be used as a potential marker that may assist in the identification of CS patients that should undergo prolonged cardiac monitoring using implantable cardiac monitors. If the present findings are externally validated, the echocardiographic and neurosonology findings may be included in current risk stratification scores (e.g., HAVOC) and other schemata for AF detection in CS [49,50].

Certain limitations of the present study need to be acknowledged. First, the sample size of the present single center study was moderate (*n* = 373). Second, there was no core laboratory analysis of CDU/TCD recordings for arrhythmia detection and no central adjudication of neuroimaging parameters. However, considering that investigators evaluating neuroimaging and neurosonology studies were blinded to the AF status of each patient, it is unlikely that this may have led to significant bias. Third, ECG detection of AF in CS patients was assessed by repetitive short-term (24-h) external monitoring devices in an outpatient setting and such an intermittent monitoring strategy has lower sensitivity and lower negative predictive value than continuous arrhythmia monitoring. Moreover, patients were not under continuous ECG monitoring during hospitalization, since the policy of our institution did not allow prolonged cardiac monitoring with repeated 24-h Holter-ECG recordings or cardiac telemetry or implantable cardiac monitoring during hospitalization. Fourth, the optimal duration of CDU/TCD recording for arrhythmia detection was not assessed in our study. It may be postulated that a more prolonged recording, for example continuous 1-h TCD monitoring using a headframe in search of arrhythmia and microembolic signals as well, could have identified more episodes of rhythm abnormalities, making the correlation with AF detection on ECG recordings even stronger. Finally, data about other possible confounders, such as secondary prevention therapies or patients’ body mass index (BMI) were not collected.

## 5. Conclusions

In conclusion, to the best of our knowledge, this is the first study demonstrating an independent association between arrhythmia detection during neurosonology examinations in the early stages of IS and the detection of AF on outpatient 24-h Holter-ECG recordings in CS patients. Our findings appear to expand the utility of CDU/TCD studies in determining stroke etiology. However, our study was not designed to evaluate the diagnostic utility of neurosonology in comparison to outpatient prolonged cardiac monitoring. Further external validation of the present findings in larger cohorts of patients with more extensive duration of cardiac monitoring is required.

## Figures and Tables

**Figure 1 jcm-08-02002-f001:**
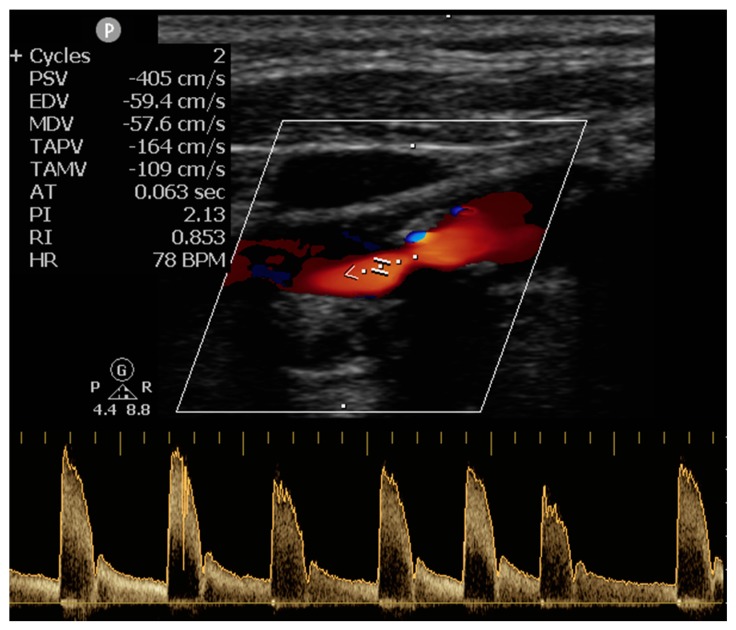
Detection of cardiac arrhythmia during spectral waveform analysis of external carotid artery in cervical duplex ultrasound.

**Table 1 jcm-08-02002-t001:** Baseline characteristics of the study population (*n* = 373).

Variable	Overall
Age, years (mean ± SD)	60 ± 11
Female sex (%)	122 (33%)
NIHSS-Score, points (median, IQR)	4 (3–10)
Hypertension (%)	230 (62%)
Diabetes (%)	82 (22%)
Hyperlipidemia (%)	215 (58%)
Current Smoking (%)	158 (22.5%)
Coronary Artery Disease (%)	58 (16%)
Excessive Alcohol Intake (%)	37 (10%)
Previous History of TIA or Stroke (%)	74 (20%)
Heart Failure (%)	17 (5%)
Peripheral Arterial Disease (%)	15 (4%)
Vascular Disease (%)	70 (19%)
CHA2DS2-VASc Score, Points (mean ± SD)	3.8 ± 1.3
Left Atrial Enlargement (%)	155 (42%)
Mild	133 (36%)
Moderate	17 (5%)
Severe	5 (1%)
Cortical Location of Infarction (%)	76 (20%)
Cardiac Arrhythmia Detected during Neurosonology Evaluation (%)	66 (18%)
Number of 24-h Holter Recordings (mean ± SD)	1.5 ± 1.5
1	254 (68%)
2	85 (23%)
≥3	34 (9%)

IQR: interquartile range, TIA: transient ischemic attack.

**Table 2 jcm-08-02002-t002:** Prevalence of atrial fibrillation detection on 24-h Holter monitoring stratified by degree of left atrial enlargement.

Left Atrial Enlargement	Atrial Fibrillation (−)	Atrial Fibrillation (+)	*p*-Value *	*p*-Value for Linear Trend **
None (%)	93%	7%	<0.001	<0.001
Mild (%)	88%	12%		
Moderate or Severe (%)	64%	36%		

* Pearson chi-square: 17.952 (df = 2); ** Linear by linear association: 12.887 (df = 1).

**Table 3 jcm-08-02002-t003:** Prevalence of atrial fibrillation detection on 24-h Holter monitoring stratified by the number of 24-h Holter-ECG recordings.

Number of 24-h Holter ECG Recordings	Atrial Fibrillation (−)	Atrial Fibrillation (+)	*p* Value *	*p*-Value for Linear Trend **
1 (%)	92%	8%	0.048	0.014
2 (%)	86%	14%		
≥3 (%)	79%	21%		

* Pearson chi-square: 6.079 (df = 2); ** Linear by linear association: 6.057 (df = 1).

**Table 4 jcm-08-02002-t004:** Univariable and multivariable logistic regression analyses depicting the associations of baseline characteristics with the likelihood of atrial fibrillation detection during 24-h Holter monitoring.

	Univariable Logistic Regression Analysis		Multivariable Logistic Regression Analysis	
Variable	Odds Ratio (95%CI)	*p* *	Odds Ratio (95%CI)	*p*
Age (per 10-year increase)	1.81 (1.31–2.50)	<0.001	1.68 (1.19–2.37)	0.003
Female Sex	1.43 (0.73–2.80)	0.300		
NIHSS-Score at Admission (per 1-point increase)	0.97 (0.91–1.03)	0.295		
Hypertension	1.51 (0.74–3.08)	0.254		
Diabetes Mellitus	1.40 (0.67–2.94)	0.374		
Hyperlipidemia	1.60 (0.80–3.21)	0.185		
Previous History of TIA or Stroke	1.20 (0.54–2.64)	0.655		
Coronary Artery Disease	1.17 (0.49–2.80)	0.719		
Congestive Heart Failure	2.74 (0.85–8.83)	0.093	2.74 (0.85–8.83)	0.165
Current Smoking	1.22 (0.63–2.35)	0.558		
Excessive Alcohol Intake	1.34 (0.49–3.67)	0.565		
Peripheral Arterial Disease	1.30 (0.28–5.96)	0.739		
Vascular Disease	1.09 (0.48–2.49)	0.833		
CHA2DS2-VASc Score (per 1-point increase)	1.53 (1.20–1.94)	0.001	1.15 (0.80–1.66)	0.451
≥3 (24-h) Holter Evaluations	2.40 (0.97–5.95)	0.058	1.62 (0.58–4.52)	0.354
Cortical Location of Infarction	1.35 (0.63–2.90)	0.444		
Cardiac Arrhythmia Detected during Neurosonology Evaluation	3.77 (1.87–7.60)	<0.001	3.09 (1.47–6.48)	0.003
Moderate or Severe Left Atrial Enlargement	5.70 (2.22–14.61)	<0.001	4.81 (1.77–13.03)	0.002

* cutoff of *p* < 0.1 was used for selection of candidate variables for inclusion in multivariable logistic regression models.

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
