# Peer review of "Potential Utility of Neurosonology in Paroxysmal Atrial Fibrillation Detection in Patients with Cryptogenic Stroke"

_jcm, 2019, doi:10.3390/jcm8112002_

Round 1

Reviewer 1 Report

This study evaluates the diagnostic utility of neurosonology examinations in the early detection of AF in patients with cryptogenic stroke and sinus rhythm on baseline cardiac evaluation. The authors concluded that neurosonology examinations in the early stages of ischemic stroke could improve the detection rate of AF in patients with cryptogenic stroke. This paper is interesting; however, the results presented by the authors do not fully support the authors' conclusions.

What kind of cardiac arrhythmia was detected during neurosonology evaluation? The authors should clarify the type of cardiac arrhythmia found during neurosonology evaluation. Moreover, was cardiac arrhythmia identified by the neurosonology test not detected on 24-hour Holter-ECG recordings during hospitalization?

Can patients with cardiac arrhythmia detected on the neurosonology test be classified as a cryptogenic stroke? Has this patient not undergone EKG monitoring during hospitalization? If a cardiac arrhythmia has been identified in neurosonology examinations, additional arrhythmia tests for accurate diagnosis of arrhythmia during hospitalization should be performed.

Evaluating the diagnostic value of neurosonology examinations requires comparison with a previously recommended diagnostic test for the detection of AF. Currently, prolonged outpatient cardiac monitoring is recommended for patients whose ischemic strokes are cryptogenic after conventional inpatient evaluation. What was the result of prolonged outpatient cardiac monitoring? How does the result compare to neurosonology examinations?

What was the result of other evaluation for a cardiac structure such as TEE, cardiac CT, or cardiac MRI? Why did some patients have additional 24-hour Holter-ECG recordings after discharge? I am wondering if the authors missed some critical confounders.

This paper does not appear to be a study of the potential utility of neurosonology in improving the detection rate of PAF in patients with CS, but rather a study of the independent predictors for the detection of occult AF in patients with CS. It would be better to describe it in a way that emphasizes the topic of the study.

Author Response

Response to Reviewer 1 Comments

General Comment: This study evaluates the diagnostic utility of neurosonology examinations in the early detection of AF in patients with cryptogenic stroke and sinus rhythm on baseline cardiac evaluation. The authors concluded that neurosonology examinations in the early stages of ischemic stroke could improve the detection rate of AF in patients with cryptogenic stroke. This paper is interesting; however, the results presented by the authors do not fully support the authors' conclusions.

Response: We would like to thank Reviewer #1 for reviewing our manuscript and for her/his concise summary of our study findings. No changes were made regarding the aforementioned general comment.

Point 1: What kind of cardiac arrhythmia was detected during neurosonology evaluation? The authors should clarify the type of cardiac arrhythmia found during neurosonology evaluation. Moreover, was cardiac arrhythmia identified by the neurosonology test not detected on 24-hour Holter-ECG recordings during hospitalization?

Response 1: We would like to thank Reviewer #1 for this attentive comment. We have followed her/his suggestions and added the clarification of cardiac arrythmia detected by neurosonology examinations. More specifically, we have added the following statement in the revised version of our manuscript: “Irregular duration of the intervals between consecutive peak-systolic velocities during spectral waveform analysis of extra- or intra-cranial arteries in at least 3 complexes indicated the presence of cardiac arrythmia and this finding was prospectively documented” (lines 98-101). The exact type of cardiac arrythmia cannot be identified by neurosonology evaluation and this issue is already addressed as a potential limitation in the Discussion of our original manuscript (lines 243-245). Sinus rhythm on 24-hour Holter-ECG recordings during hospitalization was a prerequisite for patients’ inclusion in our study (lines 78-79) and for the diagnosis of cryptogenic stroke. This clarification has been added in the revised manuscript: “It should be emphasized that sinus rhythm on 24-hour Holter ECG was a prerequisite for patients’ inclusion in our study and for the diagnosis of CS [3].” (lines 110-111).    

Point 2: Can patients with cardiac arrhythmia detected on the neurosonology test be classified as a cryptogenic stroke? Has this patient not undergone EKG monitoring during hospitalization? If a cardiac arrhythmia has been identified in neurosonology examinations, additional arrhythmia tests for accurate diagnosis of arrhythmia during hospitalization should be performed.

Response 2: We would like to thank Reviewer #1 for another insightful comment. Classification of ischemic strokes was perfomed using the TOAST classification (line 81). Cardiac arrythmia detection on neurosonological evaluation cannot document the presence of atrial fibrillation (lines 243-245) and the stroke should be still classified as cryptogenic. Repetition of 24-hour Holter recording can detect AF at a higher rate and is acceptable as additional arrhythmia tests for accurate diagnosis of AF in an outpatient setting. Patients were not under continuous ECG monitoring during hospitalization and this fact is added as a methodological shortcoming in our revised manuscript: “Moreover, patients were not under continuous ECG monitoring during hospitalization.” (line 264).    

Point 3: Evaluating the diagnostic value of neurosonology examinations requires comparison with a previously recommended diagnostic test for the detection of AF. Currently, prolonged outpatient cardiac monitoring is recommended for patients whose ischemic strokes are cryptogenic after conventional inpatient evaluation. What was the result of prolonged outpatient cardiac monitoring? How does the result compare to neurosonology examinations?

Response 3: We would like to thank Reviewer #1 for the aforementioned comment. AF detection during neurosonology examinations was associated with AF detection on subsequent outpatient 24-hour Holter-ECG recordings. However, our study was not designed to evaluate the diagnostic utility of neurosonology in comparison to outpatient prolonged cardiac monitoring. This limitation was included in the revised Discussion of our manuscript (lines 274-276).

Point 4: What was the result of other evaluation for a cardiac structure such as TEE, cardiac CT, or cardiac MRI? Why did some patients have additional 24-hour Holter-ECG recordings after discharge? I am wondering if the authors missed some critical confounders.

Response 4: We would like to thank Reviewer #1 for another excellent comment. Patients with CS that were included in our study had undergone TEE in 199 patients (53%). Cardiac CT and/or cardiac MRI were performed in 3 cases only, since these two investigations were not readily available in our hospital. All TEE, cardiac CT and cardiac MRI investigations did not disclose any cardiogenic source of embolization in our cohort. This information has been included in our revised manuscript: “Further evaluation regarding the cardiac structure was also conducted in a subset of our patients. Fifty-three percent of our patients (199/373) had undergone transesophageal echocardiogram (TEE). Cardiac CT and/or cardiac MRI were performed in 3 cases only, since these two investigations were not readily available in our hospital. All TEE, cardiac CT and cardiac MRI investigations did not disclose any cardiogenic source of embolization in our cohort.”  (lines 162-166).

Point 5: This paper does not appear to be a study of the potential utility of neurosonology in improving the detection rate of PAF in patients with CS, but rather a study of the independent predictors for the detection of occult AF in patients with CS. It would be better to describe it in a way that emphasizes the topic of the study.

Response 5: We appreciate Reviewer #1 for this insightful suggestion. We agree that our study sought to identify independent predictors of AF detection in patients with CS. However, we had hypothesized that cardiac arrythmia detected during neurosonology evaluation may be associated with higher likelihood of AF detection. Consequently, we have rephrased the closing statement of our Introduction in the revised manuscript as follows: “In view of former considerations, we sought to identify independent predictors of AF detection in patients with CS and sinus rhythm on baseline cardiac evaluation [electrocardiogram (ECG) & 24-hour Holter-ECG recordings], without prior AF history. More specifically, we had hypothesized that cardiac arrythmia detected during neurosonology evaluation (CDU and TCD) may be associated with higher likelihood of AF detection.” (lines 72-76) and in our revised Abstract as well : “We sought to prospectively identify independent predictors of atrial fibrillation (AF) detection in patients with CS and sinus rhythm on baseline electrocardiogram (ECG), without prior AF history and we had hypothesized that cardiac arrythmia detection during neurosonology examinations [Carotid Duplex (CDU) and Transcranial Doppler (TCD)] may be associated with higher likelihood of AF detection.” (lines 25-29).

Reviewer 2 Report

This research paper contains a lot of valuable information about how to improve the detection of PAF in patients with CS, and also independent risk factors including increasing age, LA enlargement and cardiac arrhythmia detection during neurosonology evaluations for detection of PAF in patients with CS, it is quite interesting.

General changes:

This research will be more clear to include the data (BMI of the patients, medication like ACE inhibitors and statins) in Table 1 and Table 4, and do check whether BMI is an independent risk factor for the detection of PAF with CS during neurosonology evaluations.

Author Response

Response to Reviewer 2 Comments

General Comment: This research paper contains a lot of valuable information about how to improve the detection of PAF in patients with CS, and also independent risk factors including increasing age, LA enlargement and cardiac arrhythmia detection during neurosonology evaluations for detection of PAF in patients with CS, it is quite interesting.

Response: We would like to thank Reviewer #2 for reviewing our manuscript and her/his positive feedback. No changes were made regarding the aforementioned general comments.

Point 1: This research will be more clear to include the data (BMI of the patients, medication like ACE inhibitors and statins) in Table 1 and Table 4, and do check whether BMI is an independent risk factor for the detection of PAF with CS during neurosonology evaluations.

 Response 1: We appreciate Reviewer #2 for this helpful suggestion. Unfortunately, we did not collect data on secondary prevention therapies and BMI, because those variables have not been previously associated with AF detection in CS. However, we have acknowledged this limitation in our revised Discussion (lines 269-270).   

Reviewer 3 Report

the paper is well written and the topic is interesting.

I think there is an important bias that should be clarify: when you decide to perform ECG Holter? Was abnormal sonology a reason to perform a second ECG Holter?

Moreover abnormal sonology can represent atrial fibrillation as simple extrasistolic beats and so, in this terms, the specificity of the exam as predictor of atrial fibrillation results low.

At least regarding the topic of atrial fibrillation I would like to suggest some references that could help to improve the paper:

Lip GYH, Nieuwlaat R, Pisters R, Lane DA, Crijns HJGM, et al. Refining clinical risk stratification for predicting stroke and thromboembolism in atrial fibrillation using a novel risk factor-based approach: The Euro Heart Survey on atrial fibrillation. CHEST, vol. 137, p. 263-272, ISSN: 0012-3692, doi: 10.1378/chest.09-1584

Nieuwlaat R, Prins MH, Le Heuzey JY, Vardas PE, Aliot E, Santini M, Cobbe SM, Widdershoven JWMG, Baur LH, Lévy S, Crijns HJGM, et al. Prognosis, disease progression, and treatment of atrial fibrillation patients during 1 year: Follow-up of the Euro Heart Survey on Atrial Fibrillation. EUROPEAN HEART JOURNAL, vol. 29, p. 1181-1189, ISSN: 0195-668X, doi: 10.1093/eurheartj/ehn139

Novo G, Baiamonte V, Nuccio A, Fazio G, Corrado E, Coppola G, Muratori I, Vernuccio L, Novo S. Atrial fibrillation and mild cognitive impairment: what correlation? Puccio D, Minerva Cardioangiol. 2009 Apr;57(2):143-50

Coppola G, Manno G, Mignano A, Luparelli M, Zarcone A, Novo G, Corrado E. Management of Direct Oral Anticoagulants in Patients with Atrial Fibrillation Undergoing Cardioversion.
Medicina (Kaunas). 2019 Sep 30;55(10).

Author Response

Response to Reviewer 3 Comments

General Comment: the paper is well written and the topic is interesting.

Response: We would like to thank Reviewer #3 for reviewing our manuscript and her/his supportive comments. No changes were made regarding the aforementioned general comments.

Point 1: I think there is an important bias that should be clarify: when you decide to perform ECG Holter? Was abnormal sonology a reason to perform a second ECG Holter?

Response 1: We appreciate Reviewer #3 for this constructive comment. We wish to clarify that the decision to perform a second ECG Holter was based on the presence of premature atrial contractions in the first ECG Holter and was not related to the neurosonology findings. We have added the following statement in our revised Methods: “CS patients underwent ≥1 outpatient 24-hour Holter ECG recordings, based on the presence of premature atrial contractions on the baseline 24-hour Holter ECG and that decision was not related to the neurosonology findings.” (lines 112-114).         

Point 2: Moreover abnormal sonology can represent atrial fibrillation as simple extrasistolic beats and so, in this terms, the specificity of the exam as predictor of atrial fibrillation results low.

Response 2: We really agree with Reviewer #3 that the specificity of neurosonology is low and we have added the following statement in our revised Discussion: “Abnormal neurosonology examination can represent AF as simple extra-systolic beats and consequently the specificity of this examination as a predictor of AF appears low.” (lines 245-247).

Point 3: At least regarding the topic of atrial fibrillation I would like to suggest some references that could help to improve the paper:

Lip GYH, Nieuwlaat R, Pisters R, Lane DA, Crijns HJGM, et al. Refining clinical risk stratification for predicting stroke and thromboembolism in atrial fibrillation using a novel risk factor-based approach: The Euro Heart Survey on atrial fibrillation. CHEST, vol. 137, p. 263-272, ISSN: 0012-3692, doi: 10.1378/chest.09-1584

Nieuwlaat R, Prins MH, Le Heuzey JY, Vardas PE, Aliot E, Santini M, Cobbe SM, Widdershoven JWMG, Baur LH, Lévy S, Crijns HJGM, et al. Prognosis, disease progression, and treatment of atrial fibrillation patients during 1 year: Follow-up of the Euro Heart Survey on Atrial Fibrillation. EUROPEAN HEART JOURNAL, vol. 29, p. 1181-1189, ISSN: 0195-668X, doi: 10.1093/eurheartj/ehn139

Novo G, Baiamonte V, Nuccio A, Fazio G, Corrado E, Coppola G, Muratori I, Vernuccio L, Novo S. Atrial fibrillation and mild cognitive impairment: what correlation? Puccio D, Minerva Cardioangiol. 2009 Apr;57(2):143-50

Coppola G, Manno G, Mignano A, Luparelli M, Zarcone A, Novo G, Corrado E. Management of Direct Oral Anticoagulants in Patients with Atrial Fibrillation Undergoing Cardioversion.

Medicina (Kaunas). 2019 Sep 30;55(10).

Response 3: We would like to thank Reviewer #3 for these excellent literature suggestions. We have cited the references as appropriate (References #32-34, 49)  

Round 2

Reviewer 1 Report

Thank you for the positive responses of the author. However, the matter I am most concerned about has not been solved. The impact of the study is reduced because the cardiac arrhythmia identified by the neurosonology test was most likely paroxysmal AF (pAF).

pAF may not be detected in single 24-hour Holter-ECG recordings. The arrhythmia seen in the patient's neurosonology test was likely to be pAF. If the arrhythmia seen on ultrasound is pAF, these patients are not initially adapted to the study subject. As the authors argue, lone AF that does not last longer than 30 seconds is considered one of the major risk sources of cardioembolism in the TOAST classification.

Why has this patient not undergone EKG monitoring during hospitalization? If pAF or PSVT has been suspected in neurosonology examinations, EKG monitoring for accurate diagnosis of arrhythmia during hospitalization should be performed. Instead of giving a diagnosis of cryptogenic stroke and undergoing 24-hour Holter-ECG recordings once or twice during a few years after discharge, authors should have suspected cardioembolism and an accurate assessment during hospitalization.

Another major concern is that the number and timing of outpatient 24-hour Holter-ECG recordings vary from patient to patient. Why did some patients have 3 or more 24-hour Holter-ECG recordings after discharge? Patients who have seen arrhythmias on the neurosonology may have had 24-hour Holter-ECG recordings more often because it is suspected of having AF. As the authors have confirmed, the more often the 24-hour Holter-ECG recordings, the higher the likelihood of finding AF. Because of this selection bias, it may seem as the neurosonology test predicts AF better.

Author Response

General Comment: Thank you for the positive responses of the author. However, the matter I am most concerned about has not been solved. The impact of the study is reduced because the cardiac arrhythmia identified by the neurosonology test was most likely paroxysmal AF (pAF).

Response: This is correct. Neurosonology evaluation can assist in detection of paroxysmal AF. Furthermore, secondary prevention in cryptogenic stroke (CS) includes oral anticoagulation (OAC) when AF is detected, regardless of AF pattern (paroxysmal or chronic). Notably, the benefit of OAC therapy in secondary stroke prevention in patients with AF has been established both for chronic and intermittent AF (Van Walraven et al. Oral anticoagulants vs aspirin in nonvalvular atrial fibrillation: an individual patient meta-analysis. JAMA. 2002;288(19):2441-8.). This issue has now been clarified in our revised Discussion: “Furthermore, secondary prevention in CS includes oral anticoagulation when AF is detected, regardless of AF pattern (paroxysmal or chronic). Notably, the benefit of oral anticoagulation therapy in secondary stroke prevention in patients with AF has been established both for chronic and intermittent AF [35].” (lines 205-208).    

Point 1: pAF may not be detected in single 24-hour Holter-ECG recordings. The arrhythmia seen in the patient's neurosonology test was likely to be pAF. If the arrhythmia seen on ultrasound is pAF, these patients are not initially adapted to the study subject. As the authors argue, lone AF that does not last longer than 30 seconds is considered one of the major risk sources of cardioembolism in the TOAST classification.

Response 1: We would like to thank Reviewer #1 for the aforementioned comment. However, we respectfully disagree with Reviewer #1. Ultrasound testing is not specified to diagnose AF or other types of cardiac arrhythmia. It can be used as a risk stratification tool to detect patients with CS that may be eligible to undergo prolonged cardiac monitoring (PCM) or repeat 24-hour Holter-ECG recordings. This has been underscored in detail in our revised manuscript: “One limiting factor is that, although neurosonology examinations can detect arrhythmias, it is not possible to differentiate them among the many different types and provide a certain diagnosis of AF. Abnormal neurosonology examination can represent AF as simple extra-systolic beats and consequently the specificity of this examination as a predictor of AF appears low.” (lines 248-252) and “Consequently, arrhythmia detection by CDU/TCD can be used as a potential marker that may assist in the identification of CS patients that should undergo prolonged cardiac monitoring using implantable cardiac monitors.” (lines 256-258).

Point 2: Why has this patient not undergone EKG monitoring during hospitalization? If pAF or PSVT has been suspected in neurosonology examinations, EKG monitoring for accurate diagnosis of arrhythmia during hospitalization should be performed. Instead of giving a diagnosis of cryptogenic stroke and undergoing 24-hour Holter-ECG recordings once or twice during a few years after discharge, authors should have suspected cardioembolism and an accurate assessment during hospitalization.

Response 2: We would like to thank Reviewer #1 for this attentive comment. All our patients underwent baseline 24-hour Holter-ECG monitoring during hospitalization. The baseline 24-hour Holter-ECG recording was negative for AF in all cases. Therefore, all patients fulfilled TOAST criteria for CS. This issue has been clarified in detail in the revised version of our manuscript: “It should be emphasized that sinus rhythm on baseline 24-hour Holter ECG was a prerequisite for patients’ inclusion in our study and for the diagnosis of CS [3].” (lines 110-111). We would also like to clarify that the policy of our institution does not allow PCM with repeat 24-hour Holter-ECG recordings or cardiac telemetry or implantable cardiac monitoring during hospitalization. This elucidation has been added in the revised version of our manuscript: “Moreover, patients were not under continuous ECG monitoring during hospitalization, since the policy of our institution did not allow prolonged cardiac monitoring with repeated 24-hour Holter-ECG recordings or cardiac telemetry or implantable cardiac monitoring during hospitalization.” (lines 269-271).

Point 3: Another major concern is that the number and timing of outpatient 24-hour Holter-ECG recordings vary from patient to patient. Why did some patients have 3 or more 24-hour Holter-ECG recordings after discharge? Patients who have seen arrhythmias on the neurosonology may have had 24-hour Holter-ECG recordings more often because it is suspected of having AF. As the authors have confirmed, the more often the 24-hour Holter-ECG recordings, the higher the likelihood of finding AF. Because of this selection bias, it may seem as the neurosonology test predicts AF better.

Response 3: We would like to thank Reviewer #1 for another fundamental comment. This issue has already been clarified in our revised manuscript, corresponding to a previous comment of Reviewer #3 during the first round of revision. More specifically, it should be noted that the decision to perform a second ECG Holter was based on the presence of premature atrial contractions in the first ECG Holter and was not related to the neurosonology findings. Therefore, we have added in our revised manuscript the following: “During the follow-up period that varied between 3 to 60 months, CS patients underwent ≥1 outpatient 24-hour Holter ECG recordings, based on the presence of premature atrial contractions on the baseline 24-hour Holter ECG and that decision was not related to the neurosonology findings.” (lines 111-114).

Reviewer 3 Report

Thank you for focused answer to my request. Moreover I saw that you perfected the paper with other suggested comments that make your work clearer.

Author Response

General Comment: Thank you for focused answer to my request. Moreover I saw that you perfected the paper with other suggested comments that make your work clearer.

Response: We would like to thank Reviewer #3 for reviewing our manuscript and her/his positive feedback. No changes were made regarding the aforementioned general comments.

Round 3

Reviewer 1 Report

In fact, I do not think it is a reasonable course of care to diagnose and treat patients with suspected AF on TCD for several years as a cryptogenic stroke because AF has not been confirmed on a single 24-hour Holter-ECG recording, which may not be accurate. The authors state that 24-hour Holter-ECG recording was performed after discharge regardless of the arrhythmia results seen on TCD, however, in patients with suspected AF on TCD, 24-hour Holter-ECG recording must be performed after discharge to confirm the potential of cardioembolism because patients with cryptogenic stroke and cardioembolism have a different strategy for secondary prevention.
However, I think the medical environment can be different in different countries.